# Dose Prediction and Pharmacokinetic Simulation of XZP-5610, a Small Molecule for NASH Therapy, Using Allometric Scaling and Physiologically Based Pharmacokinetic Models

**DOI:** 10.3390/ph17030369

**Published:** 2024-03-13

**Authors:** Lei Zhang, Feifei Feng, Xiaohan Wang, Hao Liang, Xueting Yao, Dongyang Liu

**Affiliations:** 1Department of Cardiology and Institute of Vascular Medicine, Peking University Third Hospital, Beijing 100191, China; zhanglei_bjmu@163.com; 2Drug Clinical Trial Center, Peking University Third Hospital, Beijing 100191, China; freda_m_ly@hotmail.com (F.F.); xiaohancpu@yeah.net (X.W.); lianghao86@126.com (H.L.); 3Center of Clinical Medical Research, Institute of Medical Innovation and Research, Peking University Third Hospital, Beijing 100191, China

**Keywords:** dose regimen, allometric scaling, physiologically based pharmacokinetic model, liver concentration prediction, XZP-5610

## Abstract

The objectives of this study were to support dose selection of a novel FXR agonist XZP-5610 in first-in-human (FIH) trials and to predict its liver concentrations in Chinese healthy adults. Key parameters for extrapolation were measured using in vitro and in vivo models. Allometric scaling methods were employed to predict human pharmacokinetics (PK) parameters and doses for FIH clinical trials. To simulate the PK profiles, a physiologically based pharmacokinetic (PBPK) model was developed using animal data and subsequently validated with clinical data. The PBPK model was employed to simulate XZP-5610 concentrations in the human liver across different dose groups. XZP-5610 exhibited high permeability, poor solubility, and extensive binding to plasma proteins. After a single intravenous or oral administration of XZP-5610, the PK parameters obtained from rats and beagle dogs were used to extrapolate human parameters, resulting in a clearance of 138 mL/min and an apparent volume of distribution of 41.8 L. The predicted maximum recommended starting dose (MRSD), minimal anticipated biological effect level (MABEL), and maximum tolerated dose (MTD) were 0.15, 2, and 3 mg, respectively. The PK profiles and parameters of XZP-5610, predicted using the PBPK model, demonstrated good consistency with the clinical data. By using allometric scaling and PBPK models, the doses, PK profile, and especially the liver concentrations were successfully predicted in the FIH study.

## 1. Introduction

Non-alcoholic fatty liver disease (NAFLD) affects approximately 1/4 of the global population, posing a significant threat to global health. Non-alcoholic steatohepatitis (NASH), which progresses from NAFLD, is often characterized by inflammation or fibrosis in conjunction with hepatic steatosis. Patients diagnosed with NASH may further develop liver cirrhosis and hepatocellular carcinoma [1,2]. Approximately 20% of individuals with NAFLD progress to NASH, with more than 40% of those subsequently developing fibrosis [3]. Moreover, NASH is often associated with a variety of metabolic disorders and contributes to a 64% increase in cardiovascular risk [4,5]. Consequently, NASH is emerging as one of the major diseases afflicting mankind.

The pathogenesis of NASH remains incompletely understood, with the “two-hit hypothesis” or “multiple-hit hypothesis” being widely recognized as the prevailing models [6]. The “two-hit hypothesis” posits that liver lipid accumulation due to obesity and insulin resistance is the initial insult, followed by a secondary hit involving inflammatory cytokines, adipokines, mitochondrial dysfunction, and oxidative stress. However, this hypothesis fails to fully capture the complexity of human NAFLD. The “multiple-hit hypothesis” suggests that multiple mechanisms, such as fat accumulation and insulin resistance, inflammatory pathways, dietary factors, genetic predisposition, etc., may work synergistically to drive the progression of the disease. To date, no therapeutic drugs for NASH have been approved for marketing in most countries worldwide. Current development in therapeutic targets for NASH mainly focus on metabolic targets, including PPARs, GLP-1R, GCGR, DPP-IV, SGLT2, FXR, and THR-β [7]. Among which, FXR is an early-developed and well-studied target [8]. FXR servers as a major intercellular bile acid (BA) receptor activated in the feeding state to regulate metabolism and inflammation [9,10,11]. The interaction between BAs and intracellular FXR not only regulates BA synthesis but also inhibits hepatic lipogenesis and steatosis, reduces hepatic gluconeogenesis, and enhances peripheral insulin sensitivity by upregulating transcription of GLUT4 [12,13]. Due to the crucial role of FXR in the progression of NASH, the development of FXR agonists for NASH treatment has become an important research direction. Numerous drugs targeting FXR are currently advancing through preclinical or clinical stages. Obeticholic acid (OCA) is the most extensively studied FXR agonist, which ameliorates inflammation and fibrosis in patients with NASH [14]. Despite the promising efficacy of FXR agonists in treating NASH, nearly all of them cause side effects such as pruritus and reduced high-density lipoprotein (HDL-C)/LDL-C ratio [15]. Some studies have suggested that the side effects of FXR agonists may be associated with target selectivity, particularly for TGR5 [16], prompting increased interest in the development of FXR-selective agonists in recent years [8]. However, due to insufficient data from approved drugs in the same class and the potential of experiencing pruritus and hepatotoxicity, there have been significant challenges in dose selection of FXR agonists in first-in-human (FIH) clinical trials. Both FDA and NMPA have issued guidelines for FIH dose design [17], recommending the use of allometric scaling (AS) and other models for dose prediction and selection. This necessitates careful selection of extrapolation species and allometric scaling models to ensure the scientific rigorousness and reliability of the results [18].

Meanwhile, FXR is primarily expressed in the intestine and liver [19], with the liver being the main site of action for FXR agonists. Consequently, it is crucial to determine hepatic drug concentration for a quantitative understanding of the pharmacological effects of FXR agonists, as the plasma concentration of FXR ligands did not reliably predict the potency of the FXR agonists [20]. Meanwhile, existing biomarkers do not effectively reflect the therapeutic effects of drugs for NASH treatment [21]. However, directly assessing liver drug concentrations in clinical practice is often invasive and challenging. Therefore, it is necessary to predict these concentrations with the help of modeling. Physiologically based pharmacokinetic (PBPK) models are widely used for predicting drug pharmacokinetics (PK) profiles. In addition, the mechanism-based modeling characteristic of PBPK models allows them to predict drug concentrations in multiple tissues, making them an appropriate tool with which to evaluate the concentrations of FXR agonists in liver.

XZP-5610, a novel non-steroidal FXR agonist developed by Xuanzhu Biopharm, has demonstrated safety and efficacy in the preclinical stage. In this study, the physicochemical properties, ADME characteristics, safety, and efficacy data were determined and employed to predict the corresponding parameters in the human and doses in FIH trails using various AS models. The obtained parameters were then used to construct PBPK models of XZP-5610 in rats and healthy Chinese volunteers. After being validated using preclinical and clinical data, the established models were employed to predict the liver concentration of XZP-5610 in humans. The PBPK model established in this research will provide useful information for the clinical design and development of XZP-5610 and other drugs in this class.

## 2. Results

### 2.1. Selection of Extrapolated Species for XZP-5610

Following co-incubating of XZP-5610 with hepatocytes from various species, a total of sixteen metabolites were identified. Among these species, mice hepatocytes displayed the highest diversity in metabolites, while the metabolic profiles of rats, dogs, and monkeys closely resembled that of humans. Therefore, for rodents, rats were a preferable choice compared to mice. Metabolic stability experiments indicated that the metabolic rates in dogs and monkeys were close to that of humans, demonstrating a moderate level of clearance. As a result, both dogs and monkeys were suitable as the extrapolated species for non-rodents. Considering the proximity to human metabolism and cost, SD rats and beagle dogs were ultimately selected as the extrapolation species for subsequent studies. The metabolites and metabolic stability results were shown in Figure 1 and Appendix A.

### 2.2. Parameters of XZP-5610 for Extrapolation and Physiologically Based Pharmacokinetic Model

#### 2.2.1. Apparent Permeability Coefficient of XZP-5610

After loading XZP-5610 at concentrations of 1, 5, and 30 µM XZP-5610, the mean P_app_ of XZP-5610 from apical to basolateral side were 5.29, 10.3, and 14.7 × 10^−6^ cm/s, respectively, while from basolateral to apical side were 7.51, 7.12, and 6.09 × 10^−6^ cm/s, respectively. The results indicated that XZP-5610 was highly permeable and may not be a substrate for transporters of gastrointestinal tract.

#### 2.2.2. Plasma Protein Binding of XZP-5610 in Different Species

At three tested concentrations, XZP-5610 showed a high protein binding ratio of >98%, with no observed species differences or dose-dependent effects. The average free fraction for CD-1 mice, SD rats, beagle dogs, cynomolgus monkeys, and humans were 0.55%, 1.3%, 0.3%, 0.75%, and 0.2%, respectively.

#### 2.2.3. Blood and Tissue Distribution Coefficients of XZP-5610 in SD Rats

The PK parameters of XZP-5610 in various tissues were individually calculated using a non-compartmental model. As shown in Appendix A, the parameters indicated that following oral administration, the C_max_ of XZP-5610 was reached at 0.5 h in most tissues, with gastrointestinal tract and liver identified as the primary distribution organs. Calculated BP ratios at 0.5, 2, and 8 h post-dosing were 0.54, 0.55, and 0.66, respectively, and the average value of 0.58 was used as the BP distribution coefficient.

In addition, the results suggested that liver was one of the target organs of XZP-5610, thus assessing and predicting hepatic concentrations were crucial. The liver-to-plasma coefficient, calculated with the same method as the BP ratio, was 12.9, and the parameter was used for the development of the PBPK model in SD rats.

#### 2.2.4. Pharmacokinetic Parameters of XZP-5610 in SD Rats and Beagle Dogs

The PK parameters of XZP-5610 in SD rats, following single intravenous or oral administration at different doses, were presented in Table 1. After a single intravenous injection of 1 mg/kg XZP-5610 to male and female SD rats, the mean AUC_0–t_ of XZP-5610 in plasma was 488.3 and 569.8 h·ng/mL, respectively. Following oral gavage at low, medium, and high doses of XZP-5610 to SD rats, rapid absorption was observed and both C_max_ and AUC showed a positive correlation with dose. The bioavailability calculated was 9.6% (2 mg/kg) and 15.6% (1 mg/kg) in male and female rats, respectively. Gender difference was found in C_max_, AUC, and F, while T_max_ and T_1/2_ were generally consistent.

The PK parameters of XZP-5610 in beagle dogs following single intravenous or oral administration at different doses were summarized in Table 2. Following a single intravenous injection of 0.2 mg/kg XZP-5610 in male and female beagle dogs, the systemic exposure AUC_0–t_ in plasma was 864.4 and 871.4 h.ng/mL, respectively. After oral gavage of different doses of XZP-5610 to male and female beagle dogs, the systemic exposure AUC_0–t_ and C_max_ in plasma demonstrated dose-dependent increases, with significantly longer T_1/2_ compared to rats. The bioavailability was 51.2% and 38.2% in male and female (0.2 mg/kg), respectively. No gender differences were observed, indicating the presence of species differences.

### 2.3. Determination of the MABEL and NOAEL Dose of XZP-5610 in Mice, Rats, and Dogs

In both acute and long-term toxicity experiments, all animals survived until the endpoint of the studies. In acute toxicity studies in rats, some dosing groups exhibited significant decreases in body weight, though these changes were not dose dependent. In long-term toxicity studies, alterations in body weight and clinical biochemistry parameters, such as ALB, A/G, ALP, TP, TCHO, HDL, and LDL, occurred at the end of the dosing period, displaying a dose-related pattern. These changes were reversible upon cessation of treatment (Appendix A). Organ examination showed dose-dependent alterations in the liver and spleen, including hepatic vacuolar-like changes at high doses, while most other parameters remained relatively unchanged. Taking into consideration of the above results, the NOAEL in rats for this study was estimated at 1.5 mg/kg for males and 1 mg/kg for females.

Acute toxicity studies in beagle dogs showed no abnormalities except for gastrointestinal reactions. In long-term toxicity studies, dose-dependent reductions in TCHO and LDL levels were observed in the mid-dose to high-dose groups, along with gastrointestinal reactions. Other symptoms were not prominent, and no significant toxicity was observed at the low dose. Therefore, the NOAEL in beagle dogs was evaluated at 0.05 mg/kg.

Various PD models were employed to assess the MABEL of XZP-5610. In the HFD-induced mice NAFLD model, doses of 1, 3, and 10 mg/kg significantly reduced NAS scores and demonstrated dose-dependent decreases in LDL, TC, TG, and HDL levels, without significant impact on liver/body weight ratios, ALT, and AST. In the STZ-DEN-HFD mice NASH model, the mid to high-dose groups exhibited decreasing TC, TG, and LDL-C levels. Moreover, at 0.3 mg/kg, a reduction in both NAS and fibrosis scores was observed. In the HFD+CCl_4_-induced rat NASH model, 0.5 mg/kg XZP-5610 demonstrated improved fibrosis and decreased the NAS score, establishing it as the minimum effective dose for rats. Considering these results, the MABEL of XZP-5610 for NASH treatment in mice was determined as 1 mg/kg, while in rats, it was 0.5 mg/kg.

### 2.4. Prediction of Human Pharmacokinetic Parameters of XZP-5610

Various AS methods were employed to predict the human CL_i.v_ and V_ss_ of XZP-5610. The results were presented in Table 3. Data from SD rats and beagle dogs were utilized in the prediction of human CL_i.v_ using five methods. However, due to the significant difference between rats and humans in in vitro hepatocytes experiments, the average CL_i.v_ calculated based on data from beagle dogs was used as the predicted human CL_i.v_, which was 138 mL/min (8.3 L/h). Various methods were employed to predict the V_ss_ in humans, and the predicted average value (41.8 L) from SD rats and beagle dogs was used as the predicted human V_ss_. Meanwhile, based on the preclinical data in SD rats and beagle dogs, the predicted human oral bioavailability was 57.4% (range: 14.7–57.4%), and the estimated value of K_a_ was 1.46 h^−1^ (range: 0.589–2.34 h^−1^).

### 2.5. Prediction of the Doses of XZP-5610 in FIH Trials

The prediction of the MRSD in humans was first performed using the body surface area method. The extrapolation of animal NOAEL doses was utilized to calculate the MTD, resulting in a range of 1.62–14.4 mg (Table 4). Accounting for a 10-fold safety factor, the estimated MRSD for humans was within the range of 0.16–1.44 mg. Additionally, the systemic exposure method was also employed to calculate the MRSD and MTD using the predicted human bioavailability (57.4%), which resulted in an estimated range of 0.28–0.71 mg and 2.77–7.07 mg, respectively (Table 5). Due to limited data from similar drugs, a conservative MRSD of 0.15 mg was selected for FIH trial.

Additionally, using the exposure at a dose of 1.0 mg/kg in mice model and 0.5 mg/kg in rat model, the human equivalent doses (HED) were predicted in a range of 2.18~2.46 mg. Meanwhile, the body surface area method was carried out to predict the minimum effective dose (estimated at 4.80 mg) using three different PD models. According to the PD results, some improvement in NAS scores and other indicators were observed in mice at a dose of 0.3 mg/kg, although the difference was not significant. Therefore, we tentatively set the MABEL in humans at about 2 mg, while setting the MTD at 3 mg.

### 2.6. Establishment and Validation of XZP-5610 PBPK Models in Rats and Healthy Chinese Population

The PBPK model of XZP-5610 in rats were constructed based on parameters determined in preclinical experiments and subsequently verified using animal pharmacokinetic data. As shown in Figure 2, all calculated predicted/observed ratios fell within the acceptable range of 0.5–2.0, with the majority of the ratios distributed within 0.7–1.5. Additionally, the predicted liver-to-plasma ratio of XZP-5610 was in good agreement with the observed data (11.8 vs. 12.9), with the K_P,L_ set as 37.0. The simulations indicated a high level of accuracy in predicting the C_max_, AUC, and tissue distribution of XZP-5610 using this PBPK model. The detailed parameters used for the rat PBPK model development were listed in Table 6.

The human PBPK model was subsequently established based on parameters from the rat PBPK model (Table 6), with f_up_, CL_i.v_, and V_ss_ determined or predicted through preclinical studies. The PK parameters and profiles across different dose groups of the single ascending dose (SAD) trail were simulated, and the obtained C_max_, CL/F, and AUC were validated using clinical data. Results were presented in Figure 3, which showed that the majority of the predicted data fell within the acceptable range of 0.8–1.25.

### 2.7. Prediction of XZP-5610 Liver Concentrations in Healthy Chinese Population Using PBPK Models

The simulated drug concentration-time profiles of XZP-5610 in liver tissue with four different doses were presented in Figure 4. Accordingly, the liver-to-plasma ratios, calculated based on plasma and tissue C_max_ and AUC, were approximately 10.3 and 13.5, respectively. These values closely aligned with the experimentally determined data in rats, indicating a good agreement between the predicted and observed liver distributions of XZP-5610.

## 3. Discussion

FXR agonists demonstrated remarkable efficacy in the treatment of NASH disease and are considered one of the most promising potential therapeutic drugs. However, obeticholic acid, one of the fastest-progressing drugs, was rejected by the FDA due to severe side effects such as pruritus, gastrointestinal, and hepatic adverse events. Although non-steroidal and highly selective FXR agonists are believed to potentially avoid these adverse events, sufficient clinical evidence is still lacking. Therefore, the rational selection of the dose in a FIH trial is crucial for protecting subjects and expediting the drug development process.

In this study, various methods were used to predict the dose for the FIH trial. Due to insufficient clinical data, a safety factor of 10 was applied in this study. Appropriate selection of extrapolation species and modeling methods is crucial for the FIH dose prediction. Generally, animals with metabolic properties close to humans are preferred as extrapolation species. When utilizing parameters from different species for human dose prediction, two assumptions should be followed to ensure the reliability of the results: (a) there were no species differences in the PK characteristics of XZP-5610 among SD rats, beagle dogs, and humans; in addition, the PK characteristics were linear within the studied dose range, and (b) XZP-5610 produced similar pharmacological activity or toxicity at the same systemic exposure in both animals and humans. For the first assumption, the results of PK experiments in rats and dogs demonstrated linear correlations of both C_max_ and AUC within the studied dose range. Additionally, metabolite profiles indicate similarities in the type and degree of metabolites across these species. Furthermore, none of these metabolites were substrates for common transporters (predicted). Therefore, it was inferred that XZP-5610 exhibited linear PK characteristics within the studied dose range, without observed species differences. Regarding the second assumption, despite species differences, FXR demonstrated similar distribution characteristics and function [22,23], suggesting that it could produce similar activity or toxicity at the equivalent exposure. As a result, a relatively accurate assessment and prediction of exposure and PK parameters of XZP-5610 in humans could be achieved by the established PBPK model. The clinical data further supported the reliability of the methods and results.

To better predict PK parameters in humans, multiple extrapolation models were selected for predicting the clearance and apparent volume of distribution, which integrated features such as single-species, multi-species, and corrections for free fraction and hepatic blood flow. In comparison to the previous literature [24,25], the model established in this study reduced random errors by introducing a limited number of parameters, thus improving prediction accuracy and providing a more reliable analytical performance. However, a significant difference in predicted human CL_i.v_ was observed when using data obtained from two species (SD rats and beagle dogs). Compared to beagle dog, the CL_i.v_ predicted with rat data was substantially higher (~2 fold). The difference may be attributed to the stronger metabolic capacity of XZP-5610 in rats, as inferred from the results of hepatocyte stability. In comparison to rats, the results from beagle dog and human hepatocytes were more similar; thus, the results predicted with beagle dog were ultimately adopted as the CL_i.v_ in humans. Similarly, the average bioavailability in beagle dog was used as a typical value for predicting human bioavailability. For V_ss_, the predicted values obtained from data in rats and dogs were similar, suggesting the V_ss_ of XZP-5610 was consistent across different species. Meanwhile, the Øie–Tozer method can be utilized to correct the distribution of drugs in intracellular and intercellular fluids. Finally, the average predicted values derived from different methods and across both species were used as the predicted V_ss_ for humans. For K_a_, as the variation in physiological factors influencing gastrointestinal absorption results in a weak correlation between K_a_ values in animals and humans, the average value obtained from various species was employed as the representative predicted human K_a_, with the highest and lowest values used as the range of human K_a_, respectively.

In rat PK and toxicological experiments, sex differences were observed in both i.v. and oral administration, suggesting that these differences may not be related to absorption. It was established that CYP3A served as the primary metabolic enzyme in the metabolism of XZP-5610 (Appendix A), and the results of metabolites analysis showed that the proportion of parent drug in plasma of female and male rats was 31.33% and 1.75%, respectively. Significant sex differences in some CYP3a subtypes in rats have been reported [26], and for some compounds, sex differences in PK parameters have been correlated with metabolic enzymes [27]. Therefore, the sex differences observed in this study may be a result of differences in metabolic enzymes. In humans, the sex differences in CYP2C9 and CYP3A activity were small [28], which indicates that the possibility of sex differences in human PK profiles is limited. However, statistical analysis of relevant data during clinical trials is still recommended.

Tissue distribution results indicated the liver was the main target organ of XZP-5610, with a liver-to-plasma ratio up to 12.9. Although XZP-5610 was not a substrate for some known liver uptake transporters such as OATP1B1 and OATP1B3, the liver-to-plasma ratio suggested that some other transporters may be involved in the hepatic uptake of XZP-5610. The increased liver concentration was beneficial for its therapeutic effect, but also an important cause of toxicity. In this study, a rat PBPK model was first established and used to predict liver concentrations. A good match between the predicted and measured values in radiolabeled experiment indicated that the model was appropriate for predicting liver concentrations in rats. The model was further extrapolated to humans using in vivo and in vitro experimental results and applied to predict human liver drug concentrations. Combined with the preclinical toxicology results, this model could provide valuable support for the design of clinical trials of XZP-5610. Furthermore, it is worth noting that the tissue distribution studies indicated a T_max_ of 2 h for XZP-5610 in the intestines, which was much longer than in other tissues. In fact, further excretion studies have revealed that more than 90% of the administered dose of XZP-5610 was excreted in bile, suggesting that enterohepatic circulation may be the primary reason for the extended T_max_ of XZP-5610.

The result suggested that the predicted PK parameters using the PBPK model, established with data from Chinese healthy individuals, aligned well with the clinical data, only with the exception for the 1 mg and 2 mg doses. At these doses, clinical results indicated that the increments in C_max_ and AUC were comparatively lower than those of dose escalation. Based on results from radioactive isotope studies in rats, the predominant distribution of XZP-5610 occurred in the gastrointestinal tract and liver, implying a potential contribution of diverse absorption mechanisms in the intestines, including passive diffusion and uptake/efflux transport, for XZP-5610. Evidence for the involvement of transporters could be verified from the Caco-2 experiments, where increased P_app_ was observed with escalating XZP-5610 concentrations, although XZP-5610 has already been proved not to be a substrate for P-gp and BCRP. To precisely elucidate these mechanisms, further experimental validations are required in subsequent studies.

NOAEL and MABEL doses in animals can assist in calculating the recommended initial dose and effective dose in humans. For the dose selection of the FIH trial, the initial dose of 0.15 mg and maximum dose of 3 mg were used in the SAD trial. According to the extrapolation from animal experiments, the equivalent effective doses in humans ranged from 2.18 mg to 4.80 mg, with most of the predicted MTD at about 3~14 mg. As a result, adverse reactions may occur at a dose of about 3 mg in humans, which is close to the upper limit of the effective dose. As discussed above, XZP-5610 accumulated in the liver, the primary pharmacological and target organ. It was unlikely to obtain a better risk–benefit ratio with a higher dose. Therefore, the maximum dose was set at 3 mg in consideration of safety issues. Fortunately, in preclinical pharmacological experiments, beneficial pathological and blood biochemical changes were observed at doses lower than the MABEL dose. This indicates that setting the efficacy dose to a lower dosage level (~2 mg) is reasonable. The design of the series doses can provide useful information for exploring the effective dose in humans.

Certainly, there are still limitations in our study. Due to the unavailability of clinical samples, the predicted hepatic concentrations by the PBPK model were not validated with clinical data. Additionally, the liver-specific enrichment implies the potential contribution of certain transporters, but unfortunately, XZP-5610 was not a substrate such as OATP1B1/3 in the in vitro test. To address these issues, tissue biopsy samples can be utilized in the subsequent clinical trials to measure the hepatic concentrations of XZP-5610. Meanwhile, more in vitro experiments are needed for the verification of transporters. Nonetheless, our research indicated that the models established in this study can effectively predict the FIH doses of XZP-5610 and its distribution in the plasma and tissues of healthy Chinese adults.

## 4. Materials and Methods

### 4.1. Reagents and Materials

Hepatocytes of CD-1 mice, Sprague-Dawley (SD) rats, beagle dogs, and humans were purchased from BioIVT (Westbury, NY, USA). Hepatocytes of cynomolgus monkeys were purchased from RILD (Exeter, UK). Blank plasma from SD rats, beagle dogs, and humans were purchased from BioIVT. Blank plasma from CD-1 mice and cynomolgus monkeys was purchased from Beijing Vital River Laboratory Animal Technology Co., Ltd. (Beijing, China), and Suzhou Xishan Zhongke Experimental Animal Co., Ltd. (Suzhou, China), respectively. SD rats and beagle dogs were purchased from Zhejiang Vital River Laboratory Animal Technology Co., Ltd. (Jiaxing, China) and Beijing Marshall Biotechnology Co., Ltd. (Beijing, China), respectively. All other reagents used were of at least analytical grade and commercially available.

### 4.2. Selection of Extrapolation Species

Extrapolation species were selected based on hepatocyte metabolic stability and metabolite profiles. For hepatocyte metabolic stability, the frozen hepatocytes from CD-1 mice, SD rats, beagle dogs, cynomolgus monkeys, and humans were co-incubated with 1 μM XZP-5610 for 0, 5, 15, 30, 60, 90, and 120 min. The half-life, clearance, and other parameters of XZP-5610 in different species of hepatocytes were calculated. Equations and data [29,30,31] used in calculation were available in the Appendix A. For hepatocyte metabolite profiling, hepatocytes from different species were thawed and co-incubated with 10 μM XZP-5610. The metabolites were identified using high-resolution mass spectrometry, and the relative abundance of metabolites was calculated.

### 4.3. Prediction of Human Pharmacokinetic Parameters Using Allometric Scaling Methods

#### 4.3.1. Determination of Pharmacokinetic Parameters for Extrapolation

To estimate the maximum recommended starting dose (MRSD) in the FIH study, PK parameters of XZP-5610 in humans were predicted, including intravenous clearance (CL_i.v_), steady-state apparent volume of distribution (V_ss_), bioavailability (F), and absorption rate constant (K_a_). These predictions were based on PK data gathered from studies involving a single intravenous or oral administration of XZP-5610 to either SD rats or beagle dogs.

A total of twenty-four SD rats or beagle dogs were randomly divided into four groups (each group containing three males and three females), and XZP-5610 was administrated by a single intravenous injection (1 mg/kg for rats and 0.2 mg/kg for dogs) in group one. For other three groups, XZP-5610 was administrated with a single oral gavage of 2, 6, and 20 mg/kg to male rats, 1, 3, and 10 mg/kg to female rats, and 0.05, 0.2, and 0.8 mg/kg to dogs, respectively. For all groups, blood was collected pre-dose and post-dose, and PK parameters were calculated using a non-compartmental model.

#### 4.3.2. Extrapolation of Pharmacokinetic Parameters

Several AS models were utilized for the prediction of PK parameters in humans. For CL_i.v_, single-species scaling (SSS), single-species allometric scaling (SSAS), two-species allometric scaling (TSAS), f_u_-corrected intercept method (FCIM), and hepatic blood flow (HBF) were used. The assumption of an average body weight of 60 kg for adult individuals in the Chinese population was employed. The average body weights of rats (0.245 kg) and beagle dogs (7.21 kg) observed in the pharmacokinetic studies were utilized as the representative body weights. The V_ss_ of XZP-5610 was calculated using Øie–Tozer and SSS methods. Detailed calculation methods and equations were shown in Appendix A.

To predict human bioavailability, data from single oral administration of XZP-5610 in rats and dogs during preclinical pharmacokinetic studies were employed. For human K_a_, one-compartment PK models were constructed using NONMEM (Version 7.2, ICON plc, Leopardstown, Ireland) with mean blood concentration data obtained after oral administration of XZP-5610 in rats and dogs, respectively. These models were employed to estimate the K_a_ values of rats and dogs, serving as the upper and lower threshold of human K_a_, respectively. Meanwhile, the average of these thresholds was used as the typical value of K_a_ in humans.

### 4.4. Prediction of the Dose Regimen of XZP-5610 in FIH Studies

#### 4.4.1. Determination of the No Observed Adverse Effect Level and Maximum Tolerant Doses in SD Rats and Beagle Dogs

SD rats and beagle dogs were used for acute and long-term toxicity studies, respectively. In each experiment, SD rats or beagle dogs were randomly divided into four groups and given XZP-5610 via oral gavages. Detailed information of the randomization and dosing were listed in Appendix A, respectively.

A single dose was administrated for acute toxicity studies followed by a 14-day observation period. For long-term toxicity studies, 4-week consecutive administration was conducted, followed by an 8-week observation period. Changes in body weight, food consumption, electrocardiogram (ECG), blood pressure, blood biochemistry, ophthalmology, and clinical characteristics were monitored before and after the XZP-5610 administration and during the observation period. At the end of the observation period, euthanasia was performed, followed by gross anatomy. Tissue histopathology was conducted in addition to routine blood tests to evaluate the maximum tolerant dose (MTD) and no observed adverse effect level (NOAEL) of XZP-5610 in SD rats and beagle dogs.

#### 4.4.2. Determination of the Minimum Anticipated Biological Effect Level of XZP-5610 in Mice and Rats

Pharmacodynamics (PD) studies of XZP-5610 were conducted using the high-fat diet (HFD)-induced NAFLD mice model, the streptozocin (STZ)-diethylnitrosamine (DEN)-HFD-induced NASH mice model, and the HFD+CCl_4_-induced NASH rat model. For the HFD-induced NAFLD mice model, C57BL/6N mice were fed a high-fat diet for 10 weeks to establish the NAFLD model. Following this, oral administration of 0.3, 1, 3, and 10 mg/kg of XZP-5610 was performed for 4 consecutive weeks. The PD of XZP-5610 was evaluated by determining the blood biochemistry, NAS scores, and mRNA levels of Bsep, Shp, and Fgf15. For the STZ-DEN-HFD-induced NASH model, neonatal mice were injected with STZ to destroy pancreatic B cells, followed by a high-fat diet with administration of cholesterol and bile salts for 8 consecutive weeks to establish the NASH model. XZP-5610 was then administrated orally (0.3, 1, 3, and 10 mg/kg) for 7 consecutive weeks, and the efficacy was evaluated by determining the blood biochemistry, the degree of lipoatrophy, and the NAS score. For the HFD+CCl_4_ induced NASH model, the SD rats were fed a high-fat diet for 13 weeks followed by XZP-5610 administration in 4 consecutive weeks, during which CCl_4_ was intraperitoneal injected twice a week. The evaluation of drug efficacy was conducted through the measurement of blood biochemical parameters, NAS and fibrosis score, and mRNA levels of Bsep, Shp, and Fgf15. The minimum anticipated biological effect level (MABEL) of XZP-5610 was estimated through a comprehensive assessment across these three models.

#### 4.4.3. Dose Prediction of XZP-5610 in Healthy Chinese Population

In accordance with the FDA (2015) and NMPA (2012) guidelines, the human MRSD, MABEL, and MTD in FIH trials were calculated using the body surface area method and the systemic exposure approach based on preclinical PK, PD, and safety data, with the assumption that the average adult body weight is 60 kg. The equations used in calculation were available in the Appendix A.

### 4.5. Establishment and Validation of Physiologically Based Pharmacokinetic Models for XZP-5610 in Rats and Humans

#### 4.5.1. Parameters Employed for the Development of Physiologically Based Pharmacokinetic Model

The establishment of PBPK models depended upon the precise determination of both drug-specific and physiological parameters, including molecular weight, pKa, LogP, solubility, apparent permeability coefficient (P_app_), fraction unbound in plasma (f_up_), whole blood-to-plasma (BP) distribution coefficient, CL_i.v_, K_a_, F, etc. The P_app_ was measured via determination of transwell permeability using Caco-2 cell monolayers. XZP-5610 was loaded on the apical or basolateral side of the transwell containing Caco-2 cells at concentrations of 1, 5, and 30 µM. After 120 min of incubation, samples were collected and the P_app_ of XZP-5610 at different concentrations were calculated.

The in vitro incubation was employed to measure the f_up_ of XZP-5610. Different species plasma solutions containing XZP-5610 were prepared, respectively, at final concentrations of 0.2, 2, and 10 μM. After equilibrium dialysis for 4 h, concentrations of XZP-5610 in plasma and buffer were determined by LC-MS/MS, and the f_up_ of XZP-5610 in different species were calculated.

The BP distribution and tissue–plasma distribution coefficients were determined using [^14^C] XZP-5610. A total of twenty-four SD rats were randomly divided into four groups (six rats each group, male: female 1:1). Each group received a single oral gavage of [^14^C] XZP-5610 (3 mg/100 μCi/kg). Blood samples were collected via cardiac puncture at 0.5, 2, 8, and 24 h post-dose, followed by tissue dissection and samplings. The radioactivity in different tissues were determined and the BP and tissue-plasma distribution coefficients were calculated.

#### 4.5.2. Establishment and Validation of Physiologically Based Pharmacokinetic Model for XZP-5610 in Rats

PK-sim (Version 11.2, Bayer Technology Services, Leverkusen, Germany) was used to develop PBPK models for XZP-5610 in SD rats. The distribution part of the model was optimized using Rodgers and Rowland method. For cellular permeability calculation, the PK-Sim standard method was employed. Parameters such as P_app_, f_up_, CL_i.v_, and BP ratio were obtained as described in Section 4.5.1. The other parameters, including pKa and LogP, were predicted according to the compound structure. The model was validated using the PK and tissue distribution results from rat, with a common accepted criterion that the ratio between predicted and observed values should fall within the range of 0.5 to 2.0.

#### 4.5.3. Establishment and Validation of Physiologically Based Pharmacokinetic Model for XZP-5610 in Healthy Chinese Adults

Most parameters for establishing and validating the human PBPK model were in accordance with results obtained in rats. The other parameters, which may differ from those of rats, were determined through in vitro and in vivo experiments. For instance, human f_up_ of XZP-5610 was determined using human plasma, and the human CL_i.v_ was predicted using AS methods. The established human PBPK models for XZP-5610 were assessed using clinically observed PK data. The validation process involved assessing the agreement between predicted and clinically observed PK profiles. Additionally, the accuracy of the models was verified by the ratios between the observed and predicted CL/F, C_max_, and AUC, as a widely accepted validation criterion, which should fall within the range of 0.5 to 2.0.

### 4.6. Prediction of XZP-5610 Concentration in Human Livers Using the Physiologically Based Pharmacokinetic Model

Following the successful establishment and validation of the XZP-5610 PBPK model in healthy Chinese population, simulations and predictions of XZP-5610 concentrations in the liver were performed at various doses in the FIH trial to assess hepatic concentration.

## 5. Conclusions

This study combined in vitro experiments, animal experiments, allometric scaling, and PBPK models to predict the dose, pharmacokinetic characteristics, and liver concentrations of XZP-5610 in FIH clinical trials. The established models could effectively predict the exposure of XZP-5610 in the blood and liver. The findings of this study will provide valuable insights for the design of future clinical trials and the dosing regimen of XZP-5610, thus facilitating its clinical development for NASH treatment.

## Figures and Tables

**Figure 1 pharmaceuticals-17-00369-f001:**
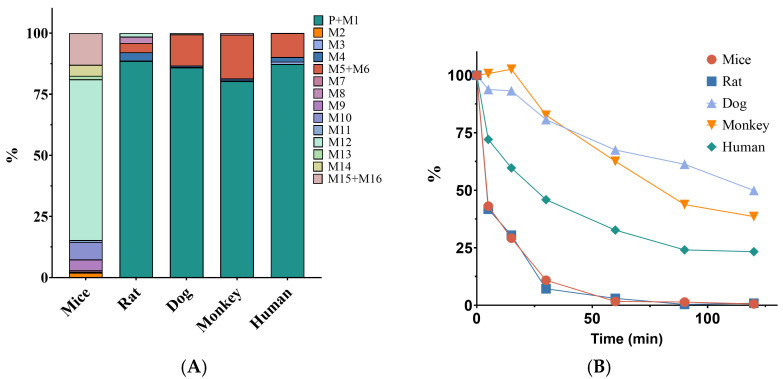
Metabolite distribution and metabolic stability of XZP-5610 in hepatocytes from different species: (**A**) metabolites identified in different species of hepatocytes; (**B**) metabolic stability of XZP-5610 in different species of hepatocytes.

**Figure 2 pharmaceuticals-17-00369-f002:**
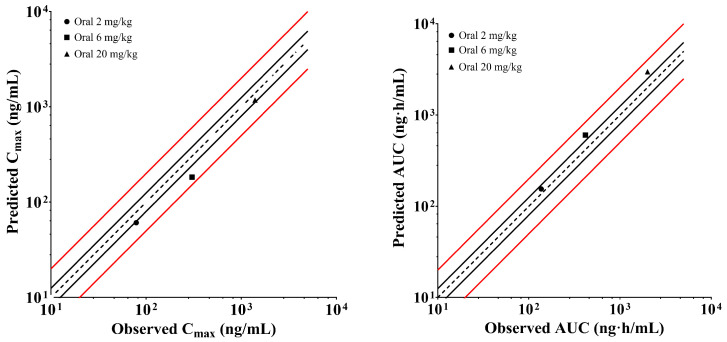
Predicted and validated pharmacokinetic results of XZP-5610 in SD rats. The black dashed line represents the predicted values corresponding to 100% of the observed values. The black solid line represents 0.8 and 1.25 times of the observed values. The red solid line represents 0.5 and 2 times of the observed values. The observed C_max_ or AUC was the average values of male and female rats with the assumption that the PK behavior was linear within the doses.

**Figure 3 pharmaceuticals-17-00369-f003:**
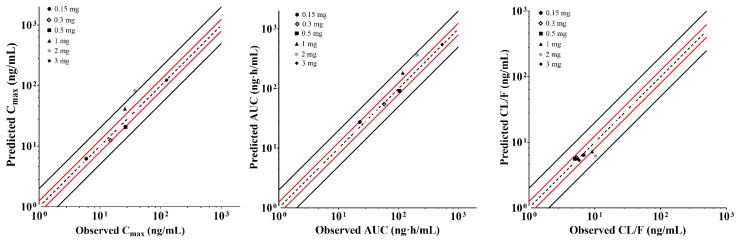
Predicted and validated pharmacokinetic results of XZP-5610 in humans. The black dashed line represents the predicted values corresponding to 100% of the observed values. The black solid line represents 0.8 and 1.25 times of the observed values. The red solid line represents 0.5 and 2 times of the observed values.

**Figure 4 pharmaceuticals-17-00369-f004:**
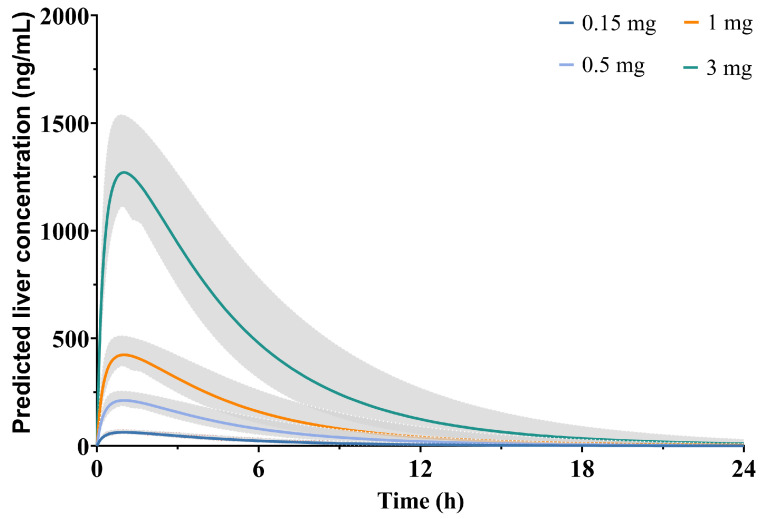
Predicted pharmacokinetic curves of XZP-5610 in human liver at doses of 0.15, 0.5, 1, and 3 mg. The dash area represents 90% confidence level.

**Table 1 pharmaceuticals-17-00369-t001:** The pharmacokinetic parameters of XZP-5610 in SD rats following single intravenous or oral administration.

Route	Dose (mg/kg)	Gender	T_max_ (h)	T_1/2_ (h)	C_max_ (ng/mL)	AUC_(0–t)_ (h·ng/mL)	Cl (L/h/kg)	V_ss_ (L/kg)	F (%)
I.V.	1	male	NA	2.9 ± 2.3	NA	488.3 ± 31.1	2.0 ± 0.1	0.9 ± 0.5	NA
I.V.	1	female	NA	2.2 ± 1.5	NA	569.8 ± 151.4	1.8 ± 0.4	0.6 ± 0.1	NA
PO	2	male	0.5 ± 0.0	1.3 ± 0.2	56.6 ± 11.6	94.9 ± 7.8	NA	NA	9.6 ± 0.8
PO	1	female	0.5 ± 0.0	1.3 ± 0.2	50.4 ± 15.0	89.0 ± 25.5	NA	NA	15.6 ± 4.5
PO	6	male	0.5 ± 0.0	1.0 ± 0.2	255.8 ± 47.7	293.8 ± 43.4	NA	NA	9.9 ± 1.5
PO	3	female	0.8 ± 0.3	1.2 ± 0.4	175.6 ± 17.9	271.1 ± 45.8	NA	NA	15.9 ± 2.7
PO	20	male	1.0 ± 0.0	1.4 ± 0.5	762.9 ± 283.0	1276.6 ± 447.0	NA	NA	12.9 ± 4.5
PO	10	female	0.5 ± 0.0	1.9 ± 0.9	1014.0 ± 544.6	1367.8 ± 341.4	NA	NA	24.1 ± 6.0

**Table 2 pharmaceuticals-17-00369-t002:** The pharmacokinetic parameters of XZP-5610 in beagle dogs following single intravenous or oral administration.

Route	Dose (mg/kg)	Gender	T_max_ (h)	T_1/2_ (h)	C_max_ (ng/mL)	AUC_(0–t)_ (h·ng/mL)	Cl (L/h/kg)	V_ss_ (L/kg)	F (%)
I.V.	0.2	Male	NA	7.9 ± 6.7	NA	864.4 ± 56.8	0.2 ± 0.03	1.1 ± 0.6	NA
		Female		5.5 ± 2.5		871.4 ± 375.8	0.2 ± 0.1	1.0 ± 0.5	
PO	0.05	Male	1.7 ± 0.6	10.1 ± 3.4	15.8 ± 5.5	88.2 ± 10.1	NA	NA	43.4 ± 4.0
		Female	1.7 ± 0.6	13.1 ± 5.0	21.1 ± 9.6	125.9 ± 63.1			82.9 ± 59.8
PO	0.2	Male	2.0 ± 1.7	6.1 ± 1.7	126.0 ± 77.5	508.3 ± 155.0	NA	NA	51.2 ± 21.1
		Female	1.3 ± 0.6	6.5 ± 1.8	67.3 ± 12.9	319.4 ± 89.6			38.2 ± 12.2
PO	0.8	Male	1.0 ± 0.0	5.8 ± 1.1	533.7 ± 277.7	2140.0 ± 1069.5	NA	NA	61.4 ± 28.9
		Female	1.3 ± 0.6	7.0 ± 2.4	626.7 ± 144.8	2279.2 ± 383.2			67.1 ± 11.1

**Table 3 pharmaceuticals-17-00369-t003:** Predicted human intravenous clearance and steady-state apparent volume of distribution with multiple allometric scaling methods.

Predicted Parameter	Predicted Method	Value
Human CL_i.v_ (mL/min)	Single-species scaling (Rat)	293
	Single-species scaling (Dog)	96.4
	Singl-species allometric scaling (Rat)	75.4
	Single-species allometric scaling (Dog)	92.3
	Two-species allometric scaling	177
	Fu-corrected intercept (Rat)	58.6
	Fu-corrected intercept (Dog)	181
	Hepatic blood flow (Rat)	845
	Hepatic blood flow (Dog)	184
Human V_ss_ (L)	Øie–Tozer	17.1
	Single-species allometric scaling (Rat)	45.3
	Single-species allometric scaling (Dog)	63.0

**Table 4 pharmaceuticals-17-00369-t004:** The predicted MRSD with the body surface area method.

Species	NOAEL Dose (mg/kg)	HED (mg)	Safety Factor	MRSD (mg)
SD rat (male)	1.5	14.4	10	1.44
SD rat (female)	1	9.60	10	0.960
Beagle dog	0.05	1.62	10	0.162

**Table 5 pharmaceuticals-17-00369-t005:** The predicted MRSD with the systemic exposure method.

Species	Gender	NOAEL Dose (mg/kg)	Steady AUC_0–24_ (ng·h/mL)	Human Equivalent AUC_0–24_ (ng·h/mL)	HED (mg)	SF	MRSD (mg)
SD rat	Male	1.5	75.2	489	7.07	10	0.71
	Female	1	46.4	302	4.36	10	0.44
Beagle dog	Male	0.05	127.6	191	2.77	10	0.28
	Female	0.05	152.1	228	3.30	10	0.33

F: the average value from PK study of beagle dog, which is 57.4%; Human CL_i.v_: Average value of the predicted human intravenous clearance, which is 8.3 L/h; the f_up_ of SD rat is 1.3%; of beagle dog, it is 0.3%; and of human, it is 0.2%.

**Table 6 pharmaceuticals-17-00369-t006:** Summary of parameters used in the PBPK model.

Property (Units)	Values Used in the Model	Data Source	Descriptions
MW(g/mol)	556.46	-	Molecular weight
pKa (Acid)	10.13, 3.39	Predicted	Acid dissociation constant
LogP	3.2	Optimized	Lipophilicity
Solubility (μg/mL)	25	Optimized	Solubility at pH 6.0
P_app_ (×10^−6^ cm/s)	10.1	Determined	Caco-2 apparent permeability
f_up_	0.013 ^a^, 0.002 ^b^	Determined	Fraction of free drug in plasma
BP	0.58	Determined	Blood-to-plasma concentration ratio
CL (L/h/kg)	1.93 ^a^, 0.138 ^b^	Determined	Clearance
CL_B_ CL_int,u_ (L/h/kg)	0.18	Optimized	Biliary clearance
K_P,L_	37.0	Optimized	Liver-to-plasma partition coefficient
Partition coefficients	Rodgers and Rowland	Optimized	Calculation method from cell to plasma coefficients
Cellular permeabilities	PK-Sim Standard	Optimized	Permeability calculation method across cell

^a^: Rat, ^b^: Human.

## Data Availability

The data presented in this study are available on request from the corresponding author.

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
