# Peer review of "Dose Prediction and Pharmacokinetic Simulation of XZP-5610, a Small Molecule for NASH Therapy, Using Allometric Scaling and Physiologically Based Pharmacokinetic Models"

_pharmaceuticals, 2024, doi:10.3390/ph17030369_

Round 1
Reviewer 1 Report
Comments and Suggestions for Authors
Summary of the Manuscript:
Reviewed the manuscript "Dose prediction and pharmacokinetic simulation of XZP-5610, a small molecule for NASH therapy, using allometric scaling and physiological based pharmacokinetic models" by Lei Zhang et al. This manuscript presents a detailed investigation on dose prediction and pharmacokinetic simulation of XZP-5610, a novel small molecule for NASH therapy. Utilizing allometric scaling and physiologically based pharmacokinetic (PBPK) models, the study aims to support dose selection for first-in-human trials and predict liver concentrations in Chinese healthy adults. Key parameters were extrapolated from in vitro and in vivo models, and a PBPK model was developed and validated with clinical data to simulate liver concentrations of XZP-5610 across different dose groups.
Strengths:
- The manuscript employs a rigorous and comprehensive methodology combining both allometric scaling and PBPK models, enhancing the reliability of dose predictions for human trials.
- It demonstrates a successful validation of the PBPK model with clinical data, indicating the model's accuracy in predicting human pharmacokinetics.
- The study addresses a significant need in NASH therapy by focusing on a novel FXR agonist, contributing valuable data for future clinical trial designs.
Recommendations for Improvement:
- Additional clarity on the selection criteria for the extrapolation species and the rationale behind choosing specific models for the PBPK simulations could enhance the manuscript's comprehensiveness.
- Incorporating a broader range of human demographic variables in the PBPK model, such as age, gender, and ethnicity, may improve the generalizability of the findings.
- A more detailed discussion on the potential limitations of the study and the implications of these findings for the clinical development of XZP-5610 would be beneficial.
Overall: The manuscript requires minor revisions. Enhancements in clarity and detail regarding the methodology and a more thorough discussion of the study's limitations and implications would strengthen the submission. The research presents important findings that significantly contribute to the field of pharmacokinetics and NASH therapy development.
Author Response
Comments and Suggestions for Authors
Summary of the Manuscript:
Reviewed the manuscript "Dose prediction and pharmacokinetic simulation of XZP-5610, a small molecule for NASH therapy, using allometric scaling and physiological based pharmacokinetic models" by Lei Zhang et al. This manuscript presents a detailed investigation on dose prediction and pharmacokinetic simulation of XZP-5610, a novel small molecule for NASH therapy. Utilizing allometric scaling and physiologically based pharmacokinetic (PBPK) models, the study aims to support dose selection for first-in-human trials and predict liver concentrations in Chinese healthy adults. Key parameters were extrapolated from in vitro and in vivo models, and a PBPK model was developed and validated with clinical data to simulate liver concentrations of XZP-5610 across different dose groups.
Strengths:
The manuscript employs a rigorous and comprehensive methodology combining both allometric scaling and PBPK models, enhancing the reliability of dose predictions for human trials.
It demonstrates a successful validation of the PBPK model with clinical data, indicating the model's accuracy in predicting human pharmacokinetics.
The study addresses a significant need in NASH therapy by focusing on a novel FXR agonist, contributing valuable data for future clinical trial designs.
Recommendations for Improvement:
Thanks for your valuable comments. We have revised our manuscript according to your suggestions which were marked in red in the manuscript. Meanwhile, the questions were answered as follows:
Additional clarity on the selection criteria for the extrapolation species and the rationale behind choosing specific models for the PBPK simulations could enhance the manuscript's comprehensiveness.
Response: Thanks for your suggestion, we have updated our manuscript accordingly (line 422).
Incorporating a broader range of human demographic variables in the PBPK model, such as age, gender, and ethnicity, may improve the generalizability of the findings.
Response: Your suggestion is highly valuable. In order to validate the PBPK model using clinical data, our study has incorporated the demographic parameters from the current clinical trials into the established PBPK model. In the subsequent Phase II or Phase III clinical trials, we will include a larger population, particularly individuals with NASH, to enhance the applicability of the developed model.
A more detailed discussion on the potential limitations of the study and the implications of these findings for the clinical development of XZP-5610 would be beneficial.
Response: Thanks for your suggestion, we have updated our manuscript accordingly (line 519).

Reviewer 2 Report
Comments and Suggestions for Authors
The article 'Dose prediction and pharmacokinetic simulation of XZP-5610, 2 a small molecule for NASH therapy, using allometric scaling 3 and physiologically based pharmacokinetic models' by Zhang, Feng et al. demonstrates a thorough investigation into the dose prediction and pharmacokinetic simulation of XZP-5610 for NASH therapy. The methodology is sound, and the results are well-presented. The discussion critically interprets the findings, acknowledging limitations and proposing considerations for future studies. The article contributes to the understanding of FXR agonists in NASH treatment and provides a foundation for further research, and merits to be published in Pharmaceuticals after minor revision.
Given the interdisciplinary nature of Pharmaceutical Sciences and its specialists, the authors would be grateful if they could present the molecular structure of XZP-5610, even if only in supplementary materials (S. M.). This can be interesting to many readers.
Authors state that they predicted that none of the metabolites were substrates for common transporters and that CYP3A served as the primary metabolic enzyme in the me tabolism of XZP-5610. Would they include any evidence of these facts in the S. M.?
The keywords are relatively broad and lack specificity. For example, "First-in-human" is a standard term for early-phase clinical trials, but it doesn't provide information on the specific context or purpose of the study. Adding more specific keywords related to the objectives, such as "dose selection" or "pharmacokinetic modeling," could help potential readers understand the study's scope better.
The introduction introduces terms like "two-hit hypothesis" and "multiple-hit hypothesis" without providing sufficient explanation. Including brief explanations or references for these terms would help readers unfamiliar with the terminology. The transition between the general discussion on NASH and the specific focus on FXR agonists could be smoother. Clearly outlining the transition and emphasizing the relevance of FXR agonists to NASH treatment would improve the coherence of the introduction.
Did the authors validate their methods by applying them to other drugs with known pharmacokinetic parameters, either in previous studies or within the current research, to ensure the reliability and validity of their procedures?
The text mentions sixteen metabolites identified after co-incubation in hepatocytes, but it might be helpful to provide a concise summary or a list of these metabolites for easier understanding.
It should be interesting comparing the stability of XZP-5610 in water with the results shown in Fig. 1B.
The explanation of the NOAEL (No Observed Adverse Effect Level) and MABEL (Minimum Anticipated Biological Effect Level) is concise, but a brief statement about their significance or how they were determined would enhance the understanding for readers not familiar with these terms.
The validation of the PBPK model is briefly discussed, but it would be beneficial to elaborate on how well the model performed and if any limitations or challenges were encountered during the validation process.
The discussion briefly touches upon the liver as the main target organ and the potential for toxicity. Expanding on the implications of increased liver concentrations and their relationship to therapeutic effects and adverse events would provide a more comprehensive understanding for readers.
In the sentence, "The prediction of FIH dose depends on appropriate species selection and accurate parameters, with two assumptions to ensure the reliability of the results," it might be clearer to explicitly state the two assumptions immediately rather than referring to them later in the text.
The discussion should end with a concise summary of key findings, their significance, and suggestions for future research directions.
The feedback provided for each of the preceding points should be incorporated into the revised version of the paper, ensuring that the comments are not solely addressed in response to this specific reviewer but are also evident in the updated manuscript.
In the S. M. section and in the headings of tables S1 and S2 it says 'does' instead of dose.
The lines 528-537 in page 14 should be removed since correspond to sections of the template that do not exist actually in the article.
Author Response
Comments and Suggestions for Authors
Thanks for your valuable comments. We have revised our manuscript according to your suggestions which were marked in red in the manuscript. Meanwhile, the questions were answered as follows:
The article 'Dose prediction and pharmacokinetic simulation of XZP-5610, 2 a small molecule for NASH therapy, using allometric scaling 3 and physiologically based pharmacokinetic models' by Zhang, Feng et al. demonstrates a thorough investigation into the dose prediction and pharmacokinetic simulation of XZP-5610 for NASH therapy. The methodology is sound, and the results are well-presented. The discussion critically interprets the findings, acknowledging limitations and proposing considerations for future studies. The article contributes to the understanding of FXR agonists in NASH treatment and provides a foundation for further research, and merits to be published in Pharmaceuticals after minor revision.
Given the interdisciplinary nature of Pharmaceutical Sciences and its specialists, the authors would be grateful if they could present the molecular structure of XZP-5610, even if only in supplementary materials (S. M.). This can be interesting to many readers.
Response: Thanks for your suggestion. I am so sorry but the structure cannot be disclosed at the moment.
Authors state that they predicted that none of the metabolites were substrates for common transporters and that CYP3A served as the primary metabolic enzyme in the metabolism of XZP-5610. Would they include any evidence of these facts in the S. M.?
Response: Thank you very much for your suggestion. ADMET Predictor was used for predicting transporter substrates, and the results showed that these metabolites were not substrates for common transporters. The metabolic enzyme subtypes were determined experimentally, and the relevant results were added in the supplementary materials (Figure S1).
The keywords are relatively broad and lack specificity. For example, "First-in-human" is a standard term for early-phase clinical trials, but it doesn't provide information on the specific context or purpose of the study. Adding more specific keywords related to the objectives, such as "dose selection" or "pharmacokinetic modeling," could help potential readers understand the study's scope better.
Response: Thank you very much for your suggestion. We have made modifications to the keywords.
The introduction introduces terms like "two-hit hypothesis" and "multiple-hit hypothesis" without providing sufficient explanation. Including brief explanations or references for these terms would help readers unfamiliar with the terminology. The transition between the general discussion on NASH and the specific focus on FXR agonists could be smoother. Clearly outlining the transition and emphasizing the relevance of FXR agonists to NASH treatment would improve the coherence of the introduction.
Response: Thank you very much for your feedback. We have updated the manuscript based on your suggestions (line 46).
Did the authors validate their methods by applying them to other drugs with known pharmacokinetic parameters, either in previous studies or within the current research, to ensure the reliability and validity of their procedures?
Response: In this study, we constructed the PBPK models in animals and humans separately according to our procedure. The models were further verified using pharmacokinetic data specific to each species to ensure the reliability of the predicted results. Furthermore, as you mentioned, the process to establish PBPK models in this study was employed for some other compounds, and the results have been partially published [1, 2].
The text mentions sixteen metabolites identified after co-incubation in hepatocytes, but it might be helpful to provide a concise summary or a list of these metabolites for easier understanding.
It should be interesting comparing the stability of XZP-5610 in water with the results shown in Fig. 1B.
Response: Thank you for your feedback. We have added the relevant information in Table S4. In addition, we tested the stability of XZP-5610 in incubation medium without hepatocytes and found that after 2 hours of incubation, XZP-5610 remained stable (99.4% remaining). However, only 0 and 2 h data in incubation medium without hepatocytes were measured so we didn’t put the results in Fig. 1B.
The explanation of the NOAEL (No Observed Adverse Effect Level) and MABEL (Minimum Anticipated Biological Effect Level) is concise, but a brief statement about their significance or how they were determined would enhance the understanding for readers not familiar with these terms.
Response: Thank you very much for your feedback. We have updated the manuscript (line 505).
The validation of the PBPK model is briefly discussed, but it would be beneficial to elaborate on how well the model performed and if any limitations or challenges were encountered during the validation process.
Response: Your suggestion is highly valuable. We have revised the manuscript and added the limitations of PBPK validation part (line 519).
The discussion briefly touches upon the liver as the main target organ and the potential for toxicity. Expanding on the implications of increased liver concentrations and their relationship to therapeutic effects and adverse events would provide a more comprehensive understanding for readers.
Response: Thank you very much for your suggestion. The information was provided in the Introduction section (line 77).
In the sentence, "The prediction of FIH dose depends on appropriate species selection and accurate parameters, with two assumptions to ensure the reliability of the results," it might be clearer to explicitly state the two assumptions immediately rather than referring to them later in the text.
Response: Thank you very much for your suggestion. We have revised the paragraph to make it clearer. (line 425).
The discussion should end with a concise summary of key findings, their significance, and suggestions for future research directions.
Response: Thank you very much for your suggestion. We have revised the manuscript accordingly. (line 526 and Conclusion section)
The feedback provided for each of the preceding points should be incorporated into the revised version of the paper, ensuring that the comments are not solely addressed in response to this specific reviewer but are also evident in the updated manuscript.
Response: We have responded to each of your comments individually and highlighted the revised text in red in the manuscript.
In the S. M. section and in the headings of tables S1 and S2 it says 'does' instead of dose.
Response: Sorry for the mistake, we have revised the content accordingly.
The lines 528-537 in page 14 should be removed since correspond to sections of the template that do not exist actually in the article.
Response: Thanks for your reminder. We have deleted the content accordingly.
Reference
1. Zhang, M.; Lei, Z.; Yu, Z.; Yao, X.; Li, H.; Xu, M.; Liu, D., Development of a PBPK model to quantitatively understand absorption and disposition mechanism and support future clinical trials for PB-201. CPT Pharmacometrics Syst Pharmacol 2023, 12, (7), 941-952.
2. Zhang, M.; Yao, X.; Hou, Z.; Guo, X.; Tu, S.; Lei, Z.; Yu, Z.; Liu, X.; Cui, C.; Chen, X.; Shen, N.; Song, C.; Qiao, J.; Xiang, X.; Li, H.; Liu, D., Development of a Physiologically Based Pharmacokinetic Model for Hydroxychloroquine and Its Application in Dose Optimization in Specific COVID-19 Patients. Front Pharmacol 2020, 11, 585021.

Reviewer 3 Report
Comments and Suggestions for Authors
The current study is set to determine the physicochemical properties, ADME characteristics, safety, and efficacy of a novel non-steroidal FXR agonist, XZP-5610, using dose prediction and pharmacokinetics simulations for NASH therapy. The experiments performed and data analyses n are suitable for reaching the target of this study. The findings of this study are valuable for the use of XZP-5610 for the treatment of NASH. Therefore, I think this work deserves to be considered for publication after the revision of the critics made below:
1. “HFD-induced NAFLD mice model, C57BL/6N mice were fed a high-fat diet for ten weeks to establish the NAFLD model.” It is well known that one cause of NAFLD is the high carbohydrate diet. Would a high carbohydrate diet have been an option in this study?
2. A total of 16 metabolites were identified in this study in the animals studied and humans. The list of the metabolites has not been provided here. This data might be critical in explaining the efficacy and toxicity of the XZP-5610. In addition, It has been mentioned in the text that 16 metabolites were identified, but only 13 were given in Figure 1.
3. Metabolic rates of dogs and monkeys were close to that of humans but were not as close as monkeys and dogs. Would it be reasonable to conclude that the metabolic profiles of rats, dogs, and monkeys closely resembled those of humans?
4. “The metabolites and metabolic stability results were shown in Figure 1 and Table S1.” Table S1 given in the supplement is about the doses of XZP-5610 administrated to SPF SD rats, not metabolites and metabolic stability.
5. Not all Tables provided in the Supplement were cited in the text.
6. Following oral administration, Cmax of XZP-506 was reached at 2 h in the intestine while at 0.5 h in other tissues. This needs clarification.
7. Standard deviations for all parameters in Table 1 and Table 2 are very high, indicating uneven distribution of the values for each individual. Therefore, the results presented here are debatable.
8. It has been stated that some dosing groups of rats exhibited significant decreases in body weight and clinical chemistry parameters and concluded that they were unrelated to doses given and the reversible changes. No data has been provided. It would have been informative if this data had been provided.
9. It isn't apparent if clinical parameters are obtained from volunteer Chinese individuals or based on the prediction. If all data is received from the volunteer individuals, information about the volunteers must be provided in the Material and Methods section. The methods used to obtain the preclinical and clinical data are not clear.
10. It has been stated that significant sex differences in some CYP3a subtypes in rats have been reported in humans. Could this be due to the low number of cases investigated in this study? Data presented in Tables 1 and 2 show high SD values, which could be due to the sampling size.
11. A conclusion section needs to be given at the end of the discussion. It would be better to provide a conclusion stating the essential findings of this study.
Comments on the Quality of English LanguageI have detected minor English grammatical errors in the text. I suggest the manuscript be checked thoroughly by a native English speaker.
Author Response
Comments and Suggestions for Authors
Thanks for your valuable comments. We have revised our manuscript according to your suggestions which were marked in red in the manuscript. Meanwhile, the questions were answered as follows:
The current study is set to determine the physicochemical properties, ADME characteristics, safety, and efficacy of a novel non-steroidal FXR agonist, XZP-5610, using dose prediction and pharmacokinetics simulations for NASH therapy. The experiments performed and data analyses n are suitable for reaching the target of this study. The findings of this study are valuable for the use of XZP-5610 for the treatment of NASH. Therefore, I think this work deserves to be considered for publication after the revision of the critics made below:
- “HFD-induced NAFLD mice model, C57BL/6N mice were fed a high-fat diet for ten weeks to establish the NAFLD model.” It is well known that one cause of NAFLD is the high carbohydrate diet. Would a high carbohydrate diet have been an option in this study?
Response: Thank you very much for your professional advice. Giving high-fat diet is a widely recognized method to induce NAFLD model, which has been used in many published articles. Therefore, in this study, we also used this method for modeling. Indeed, as you mentioned, high carbohydrate intake can lead to NAFLD, and further exploration of the differences between high sugar and high fat on model construction may be conducted in our future study.
- A total of 16 metabolites were identified in this study in the animals studied and humans. The list of the metabolites has not been provided here. This data might be critical in explaining the efficacy and toxicity of the XZP-5610. In addition, It has been mentioned in the text that 16 metabolites were identified, but only 13 were given in Figure 1.
Response: Thank you for your feedback. We have added the corresponding information of metabolites in the supplementary materials (Table S4). Additionally, since some metabolites can’t be separated on the column and only one peak can be detected in the UV detector, the total abundance for these compounds was calculated and listed. We have updated the Figure 1 accordingly. The detailed data can be seen in Tables S4.
- Metabolic rates of dogs and monkeys were close to that of humans but were not as close as monkeys and dogs. Would it be reasonable to conclude that the metabolic profiles of rats, dogs, and monkeys closely resembled those ofhumans?
Response: I apologize for any confusion caused. We have updated the manuscript (line 249 and line 251). In fact, the metabolic levels of dogs and monkeys are closer to humans and have minimal differences. Therefore, dogs were chosen as the best extrapolated species for further studies. However, for safety evaluation purposes, it is necessary to use both rodents and non-rodents. Among rodents, the metabolic levels of rats were closer to humans compared to mice. Therefore, rats were selected as representatives of rodents in our study.
- “The metabolites and metabolic stability results were shown in Figure 1 and Table S1.” Table S1 given in the supplement is about the doses of XZP-5610 administrated to SPF SD rats, not metabolites and metabolic stability.
Response: Sorry for the mistake. The table here should be Table S3. We have updated the content accordingly.
- Not all Tables provided in the Supplement were cited in the text.
Response: Sorry for the mistake. We have updated the content accordingly.
- Following oral administration, Cmax of XZP-506 was reached at 2 h in the intestine while at 0.5 h in other tissues. This needs clarification.
Response: Your opinion is highly professional. In the tissue distribution study, we also assessed the excretion of XZP-5610. From the results, we found that bile excretion accounted for more than 90% of the administered dose. Therefore, we believe that the presence of enterohepatic circulation leads to the prolonged intestinal peak time. We have added the content to the discussion section (line 488).
- Standard deviations for all parameters in Table 1 and Table 2 are very high, indicating uneven distribution of the values for each individual. Therefore, the results presented here are debatable.
Response: Your opinion is highly professional. Due to individual differences among animals, even within the same batch, significant pharmacokinetic variations can occur. As many literature results also exhibit large standard deviation values [1, 2], it is necessary to use enough animals for pharmacokinetic studies. In this study, we selected six animals per group, which is consistent with most literature practices. Additionally, we employed validated method for sample analysis to ensure the reliability of result. Therefore, the findings of this study should be considered trustworthy. Of course, future studies should adhere to more animal samples and stricter processes for drug administration, blood collection, and analysis to enhance result consistency.
- It has been stated that some dosing groups of rats exhibited significant decreases in body weight and clinical chemistry parameters and concluded that they were unrelated to doses given and the reversible changes. No data has been provided. It would have been informative if this data had been provided.
Response: Thank you for your suggestion. For body weight, after administration of XZP-5610 in acute toxicity study, male animals in the low-dose group exhibited slightly lower body weight than the control group on Day 6, while male animals in the medium-dose group showed slightly lower body weight than the control group on Day 2. Although these changes showed statistical differences, the magnitude of the changes was small, and no dose-response relationship was observed. Therefore, these changes were not considered to have toxicological significance. Meanwhile, the body weight, ALB, A/G, ALP, TP, TCHO, HDL, and LDL changed in long-term toxicity study. However, these changes were reversible upon cessation of treatment. Due to the main objective of this section is to determine the NOAEL dose in animals, specific toxicological data were not elaborated upon. As your suggestion, the detailed information was provided in supplement (Table S6).
- It isn't apparent if clinical parameters are obtained from volunteer Chinese individuals or based on the prediction. If all data is received from the volunteer individuals, information about the volunteers must be provided in the Material and Methods section. The methods used to obtain the preclinical and clinical data are not clear.
Response: Thank you very much for your suggestion. For the construction of the PBPK model, some of the parameters were derived from predictions, some from in vitro experiments and animal experiments, and some from the results of model optimization. After the preliminary establishment and validation of the rat PBPK model, these parameters were fixed to develop human PBPK model. At the same time, some human-specific parameters such as fup and Cl were also obtained from preclinical experimental measurements. The results of clinical trials were only used to verify the accuracy of the established PBPK model. To more clearly demonstrate the sources of each parameter, we have updated Table 6.
- It has been stated that significant sex differences in some CYP3a subtypes in rats have been reported in humans. Could this be due to the low number of cases investigated in this study? Data presented in Tables 1 and 2 show high SD values, which could be due to the sampling size.
Response: Your advice is very professional. In fact, there is literature reports of gender differences in the CYP3A subfamily in rats [3], but this difference is not particularly evident in humans [4]. CYP3A may be one of the factors affecting the exposure of XZP-5610 in animals. Therefore, we analyzed female and male animals’ data separately, and the results showed that there is indeed a gender difference in rats (Table 1), but the gender difference in beagle dogs is not significant (Table 2). Larger SD values may arise from various factors. As you mentioned, sample size is an important factor. Further optimization of experimental conditions can be carried out in subsequent studies.
- A conclusion section needs to be given at the end of the discussion. It would be better to provide a conclusion stating the essential findings of this study.
Response: Thank you very much for your suggestion. We have added the conclusion section as suggested.
Comments on the Quality of English Language
I have detected minor English grammatical errors in the text. I suggest the manuscript be checked thoroughly by a native English speaker.
Response: Thank you for your suggestion. We have revised our manuscript accordingly.
Reference
1. Lyu, C.; Zhang, Y.; Zhou, W.; Zhang, S.; Kou, F.; Wei, H.; Zhang, N.; Zuo, Z., Gender-Dependent Pharmacokinetics of Veratramine in Rats: In Vivo and In Vitro Evidence. Aaps j 2016, 18, (2), 432-44.
2. Baumann, M. H.; Zolkowska, D.; Kim, I.; Scheidweiler, K. B.; Rothman, R. B.; Huestis, M. A., Effects of dose and route of administration on pharmacokinetics of (+ or -)-3,4-methylenedioxymethamphetamine in the rat. Drug Metab Dispos 2009, 37, (11), 2163-70.
3. Anakk, S.; Ku, C. Y.; Vore, M.; Strobel, H. W., Insights into gender bias: rat cytochrome P450 3A9. J Pharmacol Exp Ther 2003, 305, (2), 703-9.
4. Reid, J. M.; Kuffel, M. J.; Ruben, S. L.; Morales, J. J.; Rinehart, K. L.; Squillace, D. P.; Ames, M. M., Rat and human liver cytochrome P-450 isoform metabolism of ecteinascidin 743 does not predict gender-dependent toxicity in humans. Clin Cancer Res 2002, 8, (9), 2952-62.

Round 2
Reviewer 3 Report
Comments and Suggestions for Authors
The manuscript has been revised according to most of my comments. The answers given to some of the comments have been found satidfactory.